Spatial and temporal variations of aridity shape dung beetle assemblages towards the Sahara desert

http://orcid.org/0000-0001-6558-5730 deCastro-Arrazola Indradatta 1 2 indra@mncn.csic.es
Hortal Joaquín 1 3
http://orcid.org/0000-0002-5845-3198 Moretti Marco 4
http://orcid.org/0000-0002-9852-8318 Sánchez-Piñero Francisco 2
1 Department of Biogeography and Global Change, Museo Nacional de Ciencias Naturales (MNCN-CSIC) , Madrid , Spain
2 Departamento de Zoología, Facultad de Ciencias, Universidad de Granada , Granada , Spain
3 Department of Ecology, Instituto de Ciências Biologicas, Universidade Federal de Goiás , Goiânia , Brazil
4 Biodiversity and Conservation Biology, Swiss Federal Research Institute WSL , Birmensdorf , Switzerland
Andrew Nigel
Electronic publication date: 2018 Sep 20
Publication date: 2018
Volume: 6
Electronic Location ID: e5210
Received 2018 May 1; Accepted 2018 Jun 19
Copyright: © 2018 deCastro-Arrazola et al.
Copyright year: 2018
Copyright holder: deCastro-Arrazola et al.
License: This is an open access article distributed under the terms of the Creative Commons Attribution License, which permits unrestricted use, distribution, reproduction and adaptation in any medium and for any purpose provided that it is properly attributed. For attribution, the original author(s), title, publication source (PeerJ) and either DOI or URL of the article must be cited.
License URL: https://creativecommons.org/licenses/by/4.0/

Keywords: Aridity, Water availability, Beta diversity partitioning, Environmental filtering, Resource quality, Dung beetles, Environmental gradient, Seasonality, Species turnover, Community structure

Funding: Spanish Ministry of Science and Innovation BES-2012-054353 Spanish Ministry of Science and Innovation project SCARPO CGL2011-29317 Indradatta deCastro-Arrazola was funded by an FPI grant from the Spanish Ministry of Science and Innovation (BES-2012-054353). This work was supported by the Spanish Ministry of Science and Innovation project SCARPO (grant CGL2011-29317). The funders had no role in study design, data collection and analysis, decision to publish, or preparation of the manuscript.

==============================
Background

Assemblage responses to environmental gradients are key to understand the general principles behind the assembly and functioning of communities. The spatially and temporally uneven distribution of water availability in drylands creates strong aridity gradients. While the effects of spatial variations of aridity are relatively well known, the influence of the highly-unpredictable seasonal and inter-annual precipitations on dryland communities has been seldom addressed.

Aims

Here, we study the seasonal and inter-annual responses of dung beetle (Coleoptera, Scarabaeidae) communities to the variations of water availability along a semiarid region of the Mediterranean.

Methods

We surveyed a 400 km linear transect along a strong aridity gradient from the Mediterranean coast to the Sahara (Eastern Morocco), during four sampling campaigns: two in the wet season and two in the dry season. We measured species richness, abundance and evenness. Variations in community composition between sites, seasons and years were assessed through beta diversity partitioning of dissimiliarity metrics based on species occurrences and abundances. The effects of climate, soil, vegetation and dung availability were evaluated using Spearman-rank correlations, general linear regressions and partial least-squares generalized linear regressions for community structure, and non-metric multi-dimensional scaling, Permutational Analysis of Variance (PERMANOVA) and distance-based RDA variation partitioning for compositional variations.

Results

Dung beetle abundance and species richness showed large seasonal variations, but remained relatively similar between years. Indeed, aridity and its interaction with season and year were the strongest correlates of variations in species richness and composition. Increasing aridity resulted in decreasing species richness and an ordered replacement of species, namely the substitution of the Mediterranean fauna by desert assemblages dominated by saprophagous and generalist species both in space towards the Sahara and in the dry season.

Discussion

Our study shows that aridity determines composition in dung beetle communities, filtering species both in space and time. Besides the expected decrease in species richness, such environmental filtering promotes a shift towards generalist and saprophagous species in arid conditions, probably related to changes in resource quality along the transect and through the year. Our results highlight the importance of considering the effects of the highly-unpredictable seasonal and inter-annual variations in precipitation when studying dryland communities.

Introduction

Understanding the processes behind the geographical patterns of diversity is one of the central questions of ecology. Spatial gradients have long served as natural experiments to understand general principles in the ecology of animals, through the study of how changes in environmental factors, such as climate, influence the ecological and evolutionary processes that determine biodiversity variations from local to global scales (Pianka, 1966; Willig, Kaufman & Stevens, 2003; Sanders & Rahbek, 2012). Aridity gradients emphasizing changes in water availability, a key abiotic factor, are especially important to understand geographical variations in biodiversity in warm-temperate and tropical systems (Hawkins et al., 2003; Hawkins & Porter, 2003). However, the effects of aridity on diversity remain unclear (Rohde, 1992; Willig, Kaufman & Stevens, 2003) because the response depends on taxa and geographic regions (Polis, 1991; Wiens, Kozak & Silva, 2013). For example, scorpion and bird diversity increase with aridity in North America, but decrease in Australia (Polis, 1991). On the contrary, in ants there is a negative relationship of aridity with diversity in North America but the relationship is positive in Australian deserts (Polis, 1991) and there is no relationship in Australian savannas (Andersen, Toro & Parr, 2015). Understanding biodiversity patterns across aridity gradients is also relevant because drylands currently occupy more than 40% of the land on Earth and their extent is expected to increase in the next decades in response to climate change (Huang et al., 2016).

Strikingly, temporal variations of diversity patterns along environmental gradients have received scarce attention (Bishop et al., 2014) despite their importance in shaping community assemblages and large-scale diversity patterns (Pianka, 1966; Willis & Whittaker, 2002; White et al., 2010; Gouveia et al., 2013). However, both intra-annual (seasonality) and inter-annual environmental variations play a significant role in the origin of diversity patterns and the variations of local species pools (Pianka, 1966; Tonkin et al., 2017). Temporal variations are particularly important in Mediterranean (Agoglitta et al., 2012; Seager et al., 2014) and subtropical (Belda et al., 2014) climates, where the spatially uneven distribution of water at fine and broad scales is coupled with large variations in water availability within and between years (i.e., seasonal and inter-annual variations).

Mediterranean areas typically host strong spatial gradients of water availability, often ranging from warm-temperate to desert conditions (Safriel et al., 2018). Mediterranean climate is highly seasonal, with dry hot summers and wet cool winters that result in large differences in water availability throughout the year. Importantly, besides this overall seasonal regime, arid environments show large unpredictable variations in rainfall between seasons and years, that can sometimes be larger than the typical within-year fluctuations (Ward, 2009). These differences may result in differing diversity patterns between seasons, in particular along aridity gradients, where their extremes show distinct seasonal variations—from abundant rains in winter and scarce precipitation in summer at the more humid places to scarce rains throughout the year in the desert. Drylands respond differently to extreme precipitations and seasonal rainfall than mesic biomes, with highly variable and temporally-limited increases in plant productivity (Zeppel, Wilks & Lewis, 2014). Such extreme variations in water and resource availability drive the phenology of desert animals (Polis, 1991), and may produce strong oscillations in their communities (Tonkin et al., 2017). Despite the striking changes in community composition and structure caused by these variations in biologically available water (Seely & Louw, 1980; Polis et al., 1997; Holmgren et al., 2001), the spatial and temporal effects of aridity on the diversity of invertebrates and trophic interactions have been seldom studied (Labidi, Errouissi & Nouira, 2012; Tshikae, Davis & Scholtz, 2013a).

Dung beetles of the family Scarabaeidae feed and nest on the faeces of diverse animals, especially mammalian herbivores, showing diverse dung-exploitation strategies (Hanski & Cambefort, 1991). In the Mediterranean, species in the Aphodiinae subfamily are mainly endocoprids that feed and breed within the dung pat (although many species are known to be kleptocoprids, saprophages and root feeders; Christensen & Dobson, 1976; González-Megías & Sánchez-Piñero, 2003; Dellacasa & Dellacasa, 2006), whereas the Scarabaeinae subfamily includes paracoprid and telecoprid species that feed and breed directly below or away from the dung pat, respectively (Hanski & Cambefort, 1991). The large abundances, relatively easy-to-identify species, relative stable systematics and wide distribution of dung beetles, makes them ideal to study spatial and temporal changes in community structure (Spector, 2006). Moreover, the diversity of this group is known to respond to large-scale environmental gradients (Hortal-Muñoz, Martin-Piera & Lobo, 2000; Nunes et al., 2016), in particular to variations in water availability (Haloti et al., 2006; Labidi, Errouissi & Nouira, 2012; Tshikae, Davis & Scholtz, 2013a; Abdel-Dayem et al., 2016). In dry areas dung beetles are thought to be constrained by both their physiological water economy (Sowig, 1996; Chown, Sørensen & Terblanche, 2011) and the decrease in the availability and quality of trophic resources (Lumaret, 1995; Nichols et al., 2009). Further, Palearctic Scarabaeidae are well diversified in mesic and arid Mediterranean areas (Lumaret, 1991; Sánchez-Piñero & Ávila, 2004) and surrounding desert regions (Baraud, 1985), making them a suitable model taxa to investigate biodiversity responses to aridity gradients.

Here, we study the temporal and spatial variations of dung beetle communities along a water availability gradient in Eastern Morocco, spanning 400 km from coastal Mediterranean to desert Saharan conditions. We aim to answer three specific questions: (Q1) Are dung beetle communities of desert areas reduced subsets of communities living in less arid areas? (Q2) Are community variations along the aridity gradient stable within and between years? (Q3) Do climatic factors, soil features and resource availability contribute to shape the diversity and composition of communities along the aridity gradient? According to the water–energy hypothesis (Hawkins et al., 2003), we expect a decrease in dung beetle diversity with increasing aridity. Progressively dryer conditions will filter out the most sensitive species while selecting for highly adapted species able to cope with a harsher climate and limited resource availability, thus promoting high spatial species turnover and locally distinct species assemblages (Arakaki et al., 2011; Sánchez-Piñero et al., 2011). In addition, we expect temporal shifts in the relationship between dung beetle diversity and aridity, due to the strong variations in precipitation patterns. While in Mediterranean areas the higher water availability during the wet season will cause a stronger diversity–aridity relationship than in the dry season, desert environments will show lower seasonal species turnover but higher inter-annual shifts in assemblage composition because the temporally stochastic nature of their precipitations limits the adaptation to cope with seasonal environmental shifts (Tonkin et al., 2017).

Question 1 was investigated by analyzing species richness, abundance, evenness and community composition along the aridity gradient. To answer question 2 we analyzed variations in community composition at seasonal (between wet and dry seasons) and annual (between years) temporal scales. Finally, we assessed question 3 by analyzing the effects of climate, soil, vegetation and dung availability on dung beetle assemblages.

Materials and Methods

Study area

We surveyed a linear transect spanning approximately 400 km in east Morocco, parallel to the Algerian border (Fig. 1). The transect was placed along a strong aridity gradient, from a semiarid region at the Mediterranean coast (near Saïdia, 35°5′59″N, 2°17′15″W) towards the hyperarid Sahara desert (near Figuig, 32°6′33″N, 1°13′47″W) (File S1; Table S2), with a threefold annual rainfall gradient (from ca. 350 mm at the coastal semiarid area to 100 mm at the nearly hyperarid Saharan end of the gradient) and a Mediterranean precipitation regime characterized by summer drought and rainy season in November–March (Belda et al., 2014). The difference in rainfall between the wet and dry seasons (as exemplified by the precipitation in April, the wettest month, and September, the driest month) ranged from 45 to 19 mm at the semiarid end and showed no difference at the desert end (i.e., 13 mm in both months). However, during the dry season, the greatest rainfall (22 mm) was not registered in the coast, but in the middle of the gradient.

Figure 1 Map of the ca. 400 Km transect from a semiarid zone (mor.10, Mediterranean coast) to the desert area (mor.1, Sahara desert).

The map is constructed by overlapping an aridity layer (Trabucco & Zomer, 2009) depicted in green–yellow–brown shading and an altitude layer (Hijmans et al., 2005) converted into elevation isolines (spaced 200 altitudinal meters).

Livestock breeding is the main economic activity along the whole gradient. Sheep occur along the entire transect, while cattle appears mainly in the semiarid end of the gradient towards the Mediterranean coast. Donkeys also occur along the whole transect and dromedary herds appear in the Saharan areas, but both are scarce. The transect is dominated by a single soil type (Petric Calcisols) except the coastal area near Saïdia. This area shows coastal Gleyic Solonchak soil that differ from the dominant Petric Calcisol because of a thicker silt layer (≥50 cm depth) and higher salt concentration (Jones et al., 2013). The northern part of the transect mainly corresponds to agricultural land, with forest areas restricted to mountain ranges, while the middle and the southern parts are dominated by grazing rangelands with shrub vegetation (20–30% cover, 25 ± 13 cm height; mean ± SD).

Sampling design

Dung beetles were sampled during four campaigns held in two consecutive years: two in the wet season (April 2013 and 2014) and two in the dry season (September 2013 and 2014). These months were chosen to include the two peaks of dung beetle species richness and abundance in the Mediterranean region (as recommended by Hortal & Lobo, 2005). In each campaign we surveyed 10 sampling sites along the Moroccan road N17 from the Sahara towards the Mediterranean, separated by an interval of around 40 km (Fig. 1). Sampling and beetle collection were carried out under research permits Reference Numbers 01/2013 HCEFLCD/DLCDPN/DPRN/CFF and 01/2014 HCEFLCD/DLCDPN/DPRN/CFF issued by the Haut Commisariat aux Eaux et Forêts et à Lutte Contre la Désertification (Morocco). All sampling sites were placed at least 100 m away from the road margin. Annual precipitation was similar in both years at the three places with available meteorological stations (Saïdia and Oujda in the north and Figuig in the south; www.worldweatheronline.com), although it was about 20% higher in 2012–2013 than in 2013–2014 in all three stations.

Each sampling site was replicated twice. Replicates were placed one km apart, and consisted of five baited pitfall traps (thus, 100 traps per sampling campaign) separated 20 m one from another following a straight line (thus, 80 m from trap 1 to trap 5). Each pitfall trap consisted of a one l plastic cup (11.5 cm diameter, 14 cm depth) covered by a 2 × 2 cm mesh on top of which 300 g fresh cow dung was laid as bait (see Lobo, Martín-Piera & Veiga, 1988). To avoid any spurious effects due to differences in dung composition, fresh organic-farming cow dung was harvested from a single farm (Colmenar Viejo, central Spain) and well-mixed to obtain a homogeneous dung mass, adding water and mixing right before placing the traps to ensure adequate dung moisture levels. All traps were filled with 300 ml of a soapy preservative water solution with chloral hydrate (10 g/l) to prevent quick insect degradation due to high temperatures and fungi proliferation. Traps were active for a standard period of 72 h (Labidi, Errouissi & Nouira, 2012; Amraoui et al., 2016). All captures were immediately transferred to 96% ethanol in the field and transported to the laboratory where individuals were sorted and identified to species level.

During the sampling campaigns we gathered data on resource availability, vegetation and soil characteristics (File S1; Table S1). GIS data on annual aridity (AI = mean annual precipitation/mean annual evapotranspiration) and solar radiation were obtained from Trabucco & Zomer (2009), and mean monthly temperature, annual and monthly precipitation and altitude from Hijmans et al. (2005). Resource availability was estimated using the amount of five types of dung (sheep/goat, cow, horse/donkey, dromedary and carnivore droppings) present in the locality as a general proxy for the actual amount of fresh dung that is available for dung beetles. To do this, in each replicate in the four campaigns we conducted two perpendicular 250 m long and two m wide linear transects, each surveyed by one researcher following a standardized sampling protocol, covering a total of 1,000 m2 per replicate (similar to Lobo, Hortal & Cabrero-Sañudo, 2006). In each transect the total mass of the five types of dung mentioned above was estimated based on dung volume, according to previous measurements of dung pats of different volume in the field (González-Megías & Sánchez-Piñero, 2004). Dung availability included “cow dung” and “sheep + other dung” (all expressed in g/100 m2). Vegetation height (cm) and vegetation percentage cover were estimated using the point-quarter method every five m along a 250 m × 2 m transect at each replicate and campaign to account for seasonal and yearly variations. Finally, we extracted three arbitrarily located soil cores (four cm diameter and 30 cm depth) from each replicate. Soil samples were split into three depths (0–10, 10–20 and 20–30 cm) and kept in separate air-tight plastic bags for further laboratory analyses. From these samples, seven soil variables were measured (File S1; Table S1) accounting for structure (hardness, bulk density), water content (water field capacity) and particle size (percentage of gravel, sand, silt and clay) (see Tovar, 2015).

Statistical analyses

We assessed inventory completeness for each sampling site at each campaign using Sample Coverage (Chao & Jost, 2012) as implemented in iNEXT R package (Hsieh, Ma & Chao, 2016). Average sample coverage was 99.18%, with a minimum of 91.50% for the site located at the semiarid end of the gradient in the wet season of 2014. For this reason, species richness (S) was measured as the total number of species recorded in each sampling site at each sampling campaign. Evenness (J′) was measured using Pielou’s index, that is, Shannon H′/ln(S) (Magurran, 2004).

Variations of species richness, abundance and evenness along the aridity gradient

We analyzed the relationship between aridity and species richness, abundance (mean number of individuals/trap) and evenness (question 1) through a multivariate general linear regression (GLM) with restricted sigma parameterization. We considered aridity and its interactions with both season and year as factors, to assess their eventual influence on the relationship between community descriptors and aridity. The “Aridity*Season*Year” interaction was also included to assess whether these relationships show different seasonal patterns in the two study years. Significance levels were Bonferroni-corrected, since the same analysis was carried out for three different response variables (species richness, abundance and evenness).

Variations in community composition

To assess variations in community composition (questions 1 and 2) we used non-metric multi-dimensional scaling (NMDS; Quinn & Keough, 2002), based on the Bray–Curtis similarity index. Abundance was Hellinger standardized prior to the similarity analyses, to balance relative abundances of species and minimize the double-zero problem typical of community samples (Legendre & Gallagher, 2001). Pairwise differences in Bray–Curtis similarity at each site between years, seasons and their interaction (independent of aridity) were analyzed by means of a Permutational Analysis of Variance (PERMANOVA) (9,999 iterations) using a PAST 3.15 statistical package (Hammer, Harper & Ryan, 2001).

In addition, we used a beta diversity partitioning framework based on both presence–absence data using Sørensen’s dissimilarity index and abundance data using Bray–Curtis dissimilarity index. First, Sørensen dissimilarity was partitioned into its “true species turnover” (i.e., species replacement) and “nestedness” (i.e., species loss) components (Baselga & Orme, 2012). This partitioning of dissimilarity only accounts for presence–absence variations in the data (Baselga et al., 2013), so to consider also species abundances we followed a similar procedure. We calculated abundance-based Bray–Curtis dissimilarity index (herein B–C dissimilarities for short), partitioning it into “balanced” (i.e., substitution of individuals of one species in one site by the same number of individuals of different species in another site) and “gradient” (i.e., loss of individuals from one site to another) components (Baselga, 2013). Multiple site dissimilarity was used to calculate overall beta diversity partitioning in the turnover/nestedness and balanced/gradient components of Sørensen and B–C dissimilarities, respectively, for each season in each year (i.e., for each one of the four campaigns). To analyze whether turnover/nestedness and balanced/gradient components of Sørensen and B–C dissimilarities varied along the aridity gradient, we carried out non-parametric Spearman-rank correlations between pairwise dissimilarities and the differences in aridity between each pair of sites. We used Sperman-rank correlations because residuals from GLM models did not fit a normal distribution and were heteroscedastic. All Sørensen and B–C dissimilarities calculations were done using the R package Betapart 1.5.0 (Baselga et al., 2013). In addition, we identified the most representative species of three sections of the transect representing the semiarid end, the intermediate zone and the arid end for each year and season using the indicator value index (IndVal), calculated with the R package indicspecies (De Caceres & Legendre, 2009). Due to small sample size, a Wilcoxon signed rank test was carried out to test whether the Scarabaeinae/Aphodiinae abundance ratios were lower in the dry than in the wet season, using the R package stats.

We also used Sørensen’s and B–C dissimilarities to analyze variations in community composition between seasons (wet vs dry) and between years (2013 vs 2014) for each one of the 10 points along the transect. Then, to analyze whether these site similarities between seasons and between years were related to (continuous) variations in aridity we carried out a multivariate GLM with restricted sigma parameterization considering aridity and its interactions with both season and year.

Relationship of environmental variables with diversity and composition of communities

We evaluated the relationships of species richness, abundance and evenness with climate, soil, vegetation and dung availability (question 3) through partial least-squares generalized lineal regressions (PLS-GLR; Bastien, Vinzi & Tenenhaus, 2005). This method allows analyzing data characterized by a large number of multicollinear predictor variables by extracting a set of orthogonal components (or latent vectors) considering not only the structure of predictor variables (as provided by a PCA) but also their relationship with the response variable. Once the components were computed, a GLM with Gaussian function and identity link of the response variable on the PLS components was carried out in order to test for the significance of the components and the whole model. Abundance data were log10-transformed to meet assumptions of homoscedasticity and normality of model residuals. Because communities differed mainly between seasons (see Results below), four PLS-GLR analyses were conducted (two descriptor variables × two seasons) and the critical significance level of the models was set by Bonferroni correction (p < 0.0125). Cross-validation and corrected Akaike information criterion (AICc), due to low sample size were used to select among models including different number of components. When a model component coefficient was not significant (p > 0.0125), the next optimal model indicated by cross-validation and AICc was chosen. To identify the importance of each predictor in the model, the standardized coefficients of the final PLS-GLR model were obtained by bootstrapping (1,000 iterations).

We assessed the best combination of environmental variables explaining variations in species composition (question 3) through multivariate RDA-based variation partitioning (Borcard, Gillet & Legendre, 2011) (File S3; Fig. 1). Further, we used distance-based RDA (dbRDA) to assess the best predictors of Sørensen’s and Bray–Curtis dissimilarities between all sites in each campaign. In both cases (RDA and dbRDA) we previously forward selected environmental and spatial predictors (Moran’s Eigenvector Maps; MEMs) applying the two-step procedure proposed by Blanchet, Legendre & Borcard (2008) to select significant predictor variables. We calculated the amount of variation of the different biodiversity metrics explained by the different groups of predictors (climate, space, soil and dung, as found in File S1; Table S1) and evaluated the significance of the pure fractions (for each group of variables accounting for the variance explained by all other factors) using partial RDA for species composition and partial dbRDA for Sørensen and B–C dissimilarities.

We performed a spatial autocorrelation analysis to account for any spatially-structured unexplained variability. We did this by including the vectors obtained from a Moran’s Eigenvector Map (Borcard et al., 2004; Borcard, Gillet & Legendre, 2011) into the analyses. These vectors were calculated using R packages spdep (Bivand & Piras, 2015) and spacemakeR (Dray, 2013). PLS-GLR analyses were conducted using the R package plsRglm (Bertrand, Meyer & Maumy-Bertrand, 2014), and the multivariate variation partitioning analyses with the function varpart of R package vegan (Oksanen et al., 2016).

Results

We captured 70,326 individuals of 61 dung beetle species in the four sampling campaigns (9,627 individuals of 29 Scarabaeinae species and 60,699 individuals of 32 Aphodiinae species; see File S2). Overall, dung beetle abundance and richness were slightly higher in 2014 compared to 2013 for both seasons (Fig. 2).

Figure 2 Variations in dung beetle community descriptors along an aridity gradient from a semiarid zone to the Sahara desert during the wet and dry seasons in 2013 and 2014.

Variations in species (A, D) species richness, (B, E) log abundance and (C, F) evenness along an aridity gradient from a semiarid zone (350 mm mean annual rainfall) to the Sahara desert (100 mm) during the wet (April, blue lines) and dry (September, red lines) seasons in 2013 (A–C) and 2014 (D–F). The straight lines in the figures for species richness show the regression lines for the wet and dry season each year. Abundance and evenness did not show either linear nor unimodal fits to aridity. The range of Y axis was standardized to allow easier comparison between years.

Variations in species richness, abundance and evenness

There was a significant pattern of decreasing species richness along the gradient, but neither abundance nor evenness showed a significant relationship with aridity (Fig. 2; Table 1). The relationship between species richness and aridity strongly varied seasonally, with a higher slope of the Season*Aridity interaction in the wet than in the dry season. The significant Year*Aridity interaction indicates that the relationship between species richness and aridity varied between years, 2014 showing a steeper slope than 2013 due to the higher number of species in the semiarid end of the transect, but not in the desert areas (Table 1; Fig. 2). In fact, species richness showed a clear change only during the wet season, from low values (5–10 species) in the desert to the highest richness (15–20 species) in the semiarid zone (Fig. 2). During the dry season, species richness hardly increased from the desert to semiarid areas in 2013, although a slight increase appeared near the semiarid end of the gradient, particularly in 2014.

Table 1 Multivariate GLM on the effects of temporal (seasonal and inter-annual) variations of aridity over species richness, abundance and evenness.

Effect	Species Richness F (1, 35)	log10 Abundance F (1, 35)	log10 Evenness F (1, 35)	
Aridity	15.17**	6.39	1.43	
Season*Aridity	67.07**	2.72	38.75**	
Year*Aridity	7.57*	2.48	5.16	
Year*Season*Aridity	1.11	0.02	4.27	
Notes:

Significance levels after Bonferroni correction:

* p < 0.016;

** p < 0.0003.

Total abundance did not show a significant relationship with aridity and was very similar along the gradient in both sampling seasons (Table 1; Fig. 2). The wet season showed relatively small variations of abundances throughout the gradient, with a decrease at both ends, while there was a striking increase in abundance in the semiarid end of the gradient in the dry season (Fig. 2). Although evenness was not directly related to aridity, there was a significant Season*Aridity interaction (Table 1), indicating that evenness patterns change throughout the year along the gradient, especially because of the contrasting evenness values between the wet and the dry seasons at the semiarid end of the transect (Fig. 2).

Variations in community composition

Although there were no significant relationships between total abundance and aridity, there were large seasonal differences in the relative abundance at the subfamily level. Communities in the wet season were dominated by Scarabaeinae, which accounted for 60–90% of total abundance in most sites. This contrasts with the dry season, when the communities were almost dominated by Aphodiinae, which accounted for more than 97% of abundance in most sites. Thus, the ratio of Scarabaeinae/Aphodiinae abundances significantly differed between seasons (2013: χ2 = 10632.53, p < 0.001; 2014: χ2 = 22483.74, p < 0.001). However, these differences did not hold for richness, which showed approximately a 1:1 ratio in the number of species in both seasons.

Community composition along the aridity gradient showed a strong seasonal structure. The communities during the wet and dry seasons were clearly different, except in the most arid extreme of the gradient (specially sites 2 and 3), as shown by NMDS ordination (Fig. 3). While NMDS shows a clear spatial structure in community composition along the gradient in the wet season, in the dry season only the semiarid end of the transect (sites 8–10) shows a marked difference with all the other sites (Fig. 3). PERMANOVA results corroborate that community composition significantly differed seasonally (pseudoF = 8.255, p < 0.0001, d.f. = 1.36) while showing similar patterns in both sampling years (pseudoF = 1.341, p = 0.1968, d.f. = 1.36), with no significant interaction between both factors (pseudoF = 0.604, p = 0.805, d.f. = 1.36). Changes in community composition between years were indeed small, with total Sørensen values of 0.2 and 0.3 for the wet and dry season, respectively (Fig. 4A), and total B–C of 0.3 in the wet season and 0.2 in the dry season (Fig. 4B). Indeed, the indicator species of communities (IndVal analyses) sampled were the same in 2013 and 2014 (see Appendix S3; Table 1). In contrast, three Aphodiinae species were indicators of communities sampled during the dry season (IndVal >0.95), while seven Scarabaeinae and seven Aphodiinae species were indicators of wet season communities (see File S3; Table S4).

Figure 3 Non-metric multi-dimensional scaling (NMDS) ordination (stress = 0.12) of community composition for the different sites in the wet and the dry season of the two sampled years (2013 and 2014).

Figure 4 Temporal beta diversity (Sørensen coefficient) and Bray–Curtis dissimilarities of dung beetle communities along the aridity gradient.

(A and B) Seasonal beta diversity and Bray–Curtis dissimilarities, respectively; each bar compares communities for the same site in the wet and the dry seasons. (C and D) Inter-annual beta diversity and Bray–Curtis dissimilarities, respectively; each bar compares communities for the same site in the 2 years included in this study. Beta diversity is partitioned in two components: beta turnover (dark colors) and beta nestedness (light colors). Bray–Curtis dissimilarities are also partitioned in two components: BC gradient (dark colors) and BC balanced (light colors). The range of Y axis (showing all possible variation of beta diversity or BC, 0–1) was standardized in all plots to allow easier comparison.

Variations in community dissimilarities in space and time

Dissimilarity between wet and dry season communities (both Sørensen and B–C) significantly decreased with aridity (Fig. 4), and there was no interaction with year (Table 2). In contrast, Sørensen dissimilarity between 2013 and 2014 communities was not affected by aridity (Fig. 4), although B–C dissimilarity significantly increased with aridity with an interaction with season due to the stronger pattern occurring in the dry season (Table 2; Fig. 4). Hence, total B–C dissimilarity was low between seasons in the most arid sites, but very high towards the semiarid end of the gradient (Fig. 4B).

Table 2 Variation of seasonal (wet-dry) and inter-annual (2013–2014) dissimilarity for species occurrence (Sørensen) and abundances (Bray–Curtis) in relation to aridity and its interaction with year and season.

		Sørensen		Bray–Curtiss	
Effect	Coefficient ± S.E.	F (1, 17)	Coefficient ± S.E.	F (1, 17)	
Seasonal	
Aridity	−1.974 ± 0.406	23.63**	−3.473 ± 0.458	57.58**	
Year*Aridity	0.019 ± 0.032	0.33	0.064 ± 0.036	3.14	
Inter-annual	
Aridity	0.184 ± 0.529	0.12	1.468 ± 0.424	12.00*	
Season*Aridity	0.008 ± 0.042	0.04	0.109 ± 0.034	10.49*	
Notes:

Sørensen dissimilarity for inter-annual comparison and Bray–Curtis dissimilarity for both seasonal and inter-annual comparisons were arcsine transformed. Statistical significance levels after Bonferroni correction:

* p < 0.0025;

** p < 0.00025.

Multisite Beta diversity-partitioning showed that Sørensen dissimilarity among sites was mainly due to species turnover along the gradient in both seasons, although nestedness increases in importance in the dry season (Fig. 5A). In contrast, according to B–C dissimilarity analyses balanced changes in abundance were prevalent in both seasons, with some gradient in the wet season (Fig. 5B). This pattern was similar in both years. Interestingly, both pairwise Sørensen and B–C dissimilarities showed similar relationships with pairwise differences in aridity. The turnover and balanced components increased in relation to differences in aridity between sites, while nestedness and gradient components did not change significantly with differences in aridity (Table 3).

Figure 5 Multiple site dissimilarity of dung beetle communities along the aridity gradient based on species occurrence (Sørensen) and considering species abundances (Bray–Curtis).

Multiple site dissimilarity partitioning into turnover/nestedness or balanced/gradient components of dung beetle communities along the aridity gradient for the different seasons and years sampled based on (A) species occurrence (Sørensen) and (B) considering species abundances (Bray–Curtis).

Table 3 Spearman rank correlation coefficients for the relationship between pairwise difference in aridity between sites and the components of the partitioning of Sørensen and Bray–Curtis dissimilarities.

		Sørensen	Bray–Curtis	
Year	Season	Turnover	Nestedness	Balanced	Gradient	
2013	Wet	0.429	−0.022	0.593***	−0.135	
	Dry	0.493**	−0.210	0.487*	−0.331	
2014	Wet	0.536**	−0.208	0.683***	−0.402	
	Dry	0.590***	0.116	0.479*	−0.308	
Notes:

We used presence/absence dissimilarity (turnover and nestedness components of Sørensen index) and abundance-based dissimilarity (balanced and gradient components of Bray–Curtis index) for the different surveys. Significance values after Bonferroni correction: ms, marginally significant;

* p < 0.003;

** p < 0.0006;

*** p < 0.00006.

Relationships of environmental variables with the diversity and composition of communities

The PLS-GLR analysis shows that the relationship between environmental variables and species richness and abundance along the gradient largely differed between seasons. Results for evenness are not shown as data did not meet the premises of neither normality nor homocedasticity for the wet season and were not significant for the dry season. PLS components were only significantly related to species richness variations in the wet season and to abundance variations in the dry season (File S3; Table S5). PLS standardized coefficients indicate that variations of species richness in the wet season were positively related to precipitation (both monthly and annual rainfall) and negatively related to radiation (Fig. 6A). Differences of abundance in the dry season appeared positively related to cow dung availability (Fig. 6B).

Figure 6 Distribution of standardized coefficients of PLS-GLR models for richness and abundance using all available environmental variables.

Boxplots show standardized coefficients of PLS-GLR models for species richness in the wet season (A) and abundance in the dry season (B) of dung beetle communities along an aridity gradient from a semiarid zone (350 mm mean annual rainfall) to the Sahara desert (100 mm). Standardized coefficients were obtained by bootstrap (1,000 iterations) to identify the importance of predictor variables in the models. Significant predictor variables differ from 0.

Variation partitioning of multivariate data identified significant relationships of Sørensen and B–C dissimilarities and raw community composition with climate, space, soil and dung availability (File S3; Fig. 1). As expected, space and climate explained a large proportion of the variance in all cases, although in a few cases this latter factor only rendered significant results in partial dbRDAs. Dung availability and soil variables explained a relatively small proportion of the variance in B–C dissimilarities and community composition, mainly in both seasons of 2013. Variation of Sørensen dissimilarity in all campaigns was irregularly explained (and rarely significantly, see File S3; Fig. 1) solely by climate and space (from 71% to only 23%), with the only exception of the dry season of 2013, where dung availability also explained a marginal 4%.

Discussion

Although our results support the general expectation that water availability is a major factor structuring the diversity of communities in semiarid environments, they provide novel insights on how such relationship affects community structure through time. Even though dung beetle species richness shows a clear decrease with increasing aridity, contrary to our expectations abundance and evenness did not change along the gradient. And importantly, aridity fosters a gradual replacement of species, so the most arid areas are inhabited by distinct assemblages of dung beetle species adapted to the dry and resource-poor desert conditions, rather than poor subsets of the less arid areas (question 1). But perhaps the most striking of our results is that diversity–aridity relationships show marked seasonal differences (question 2). And further, these changes may be consistent between years, rather than stochastic. This, together with the somehow unexpected lack of predictive power of resource (dung) availability (question 3) points to a strong environmental filtering as the major process behind not only of the distribution of dung beetles along the studied gradient, but also of their phenology.

Low water availability and/or precipitation is known to limit the diversity of many organisms (Sommer et al., 2010; Maestre et al., 2015), including dung beetles (Hortal, Lobo & Martin-Piera, 2001; Tshikae, Davis & Scholtz, 2013b, 2013c). Such negative relationship is, however, inexistent or even reversed in other organisms and/or systems (Polis, 1991; Polis et al., 1997; Delsinne et al., 2010; Andersen, Toro & Parr, 2015). Water availability may determine species richness through two main mechanisms: physiological constraints (Chown, Sørensen & Terblanche, 2011)—thus following the water–energy hypothesis (Hawkins et al., 2003); and resource availability (Nichols et al., 2009; Tshikae, Davis & Scholtz, 2013c)—following the species–energy hypothesis (Wright, 1983). While precipitation was directly related to species richness in our analysis, supporting the prediction of the water–energy hypothesis, dung availability did not correlate with species richness in our study. This is consistent with the claims that many other factors not directly related to resource availability may determine large-scale diversity gradients (Currie et al., 2004; Hurlbert & Jetz, 2010). In the case of our gradient, the higher dung beetle richness at the semiarid end near the coast may be sustained by the increase in diversity of dung types provided by the appearance of cow herds (Lobo, Hortal & Cabrero-Sañudo, 2006; Tshikae, Davis & Scholtz, 2013b). Further, some Aphodiinae species are generalist saprophages (Christensen & Dobson, 1976; Dellacasa & Dellacasa, 2006; Holter, Scholtz & Stenseng, 2009), so the higher availability of detritus resources (not quantified in this study) such as leaf litter in this area of higher plant production may be also promoting a higher richness.

Further, neither dung beetle abundance nor evenness showed any significant relationship with increasing aridity towards the Sahara. This result is also in contradiction with the species–energy hypothesis, and the general argument that greater productivity can maintain more individuals and therefore viable populations of a larger number of species (Hutchinson, 1959; Brown, 1981; Wright, 1983). Instead, we found an unexpected decline in abundance at the semiarid end of the gradient in the wet season. This area was, by far, the one with higher availability of cattle dung (File S1; Table S2), the richest resource present in the whole gradient, so it is unlikely that such low numbers are due to limited resource availability. Rather, the lower abundance of dung beetles in the coastal site may be partly explained by both land use intensification caused by cropland and urban spread (Davis, Scholtz & Swemmer, 2012; Nichols et al., 2013) and/or the higher salinity of the deep and superficial layers of the soil—that may deter burying dung beetle species to nest. Further, the large increase of beetle abundance at this area during the dry season was due to a single species, the aphodiid Anomius baeticus. This saprophagous beetle feeds on plant detritus (Sánchez-Piñero & Ávila, 2004; Verdú & Galante, 2004), which enables it to have massive population outbreaks in the dry season.

Strikingly, changes in community composition along the gradient follow an ordered replacement with aridity rather than a mere loss of species, although the rate of such replacement is progressively lower towards the Sahara. Most variation in composition was due to a balanced turnover in all surveys, with both Sørensen and B–C dissimilarities increasing towards the Mediterranean coast. Despite the decrease in species richness with increasing aridity, nestedness and gradient compositional changes were much smaller, remaining constant throughout the whole gradient. Such pre-eminence of species replacement indicates that the strong filtering imposed by aridity is not limited to the progressive inability of the species adapted to Mediterranean conditions to inhabit desert areas. Rather, there is a distinct pool of dung beetle species adapted to arid Saharan conditions (e.g., Onthophagus transcaspicus, Scarabaeus aegyptiacus, Mendidius palmenticola or Calamosternus lucidus; Baraud, 1985), that progressively substitutes the semiarid elements of the communities. These species are seemingly adapted to the low–and stochastic–availability of resources, and form distinct communities compared to neighboring areas with more mesic environments (Sánchez-Piñero et al., 2011). Extreme arid conditions determine the occurrence of a highly adapted biota in desert ecosystems (Arakaki et al., 2011), usually including a high proportion of endemics (Le Houérou, 2001). Whether this pattern of pre-eminence of species replacement with increasing aridity is common in desert communities needs further investigation since no other studies partitioned beta diversity along aridity gradients before.

Importantly, although the general pattern of decrease in richness and balanced turnover along the gradient holds on for all surveys, our results also show important temporal variations. Indeed, the significance of the interactions of season and year with aridity evidence that the effect of water availability on species richness changes in time. Dung beetle faunas showed strong seasonal changes, with a steeper decline of richness with aridity in the wet season. In arid and semiarid environments this season is not only characterized by milder climate, but also by the higher abundance and quality of trophic resources (Hanski, 1987). Further, the richness–precipitation relationship was weaker in 2013 than in 2014–with no significant changes in species richness across the gradient in the dry season of the former year. These differences may be related to temporal changes in precipitation, as this factor was an important predictor of richness in spring but not in the dry season. The marked precipitation gradient along the transect in spring contrasts with the scarce difference in the amount of rain along the gradient at the end of the summer-early fall (from 46 to 13 and 22 to 13 mm of monthly precipitation, respectively). Spring precipitations allow a higher plant productivity that in turn results in more hydrated dung of better quality for nesting (Lumaret, 1995), and ultimately higher reproductive success and the emergence of larger populations in the next generation. Indeed, annual precipitation was higher in 2013 than in 2012, allowing a higher abundance of dung beetles emerging the year after because of the higher reproductive success in 2013. Long-term data would be strongly needed to analyze these temporal changes in species richness and precipitation.

The spatial structure of dung beetle assemblages along the transect also varied in time. During the favorable conditions of the wet season species composition followed a structured sequence of replacement from the semiarid sites to the desert. This structure was disassembled in the harsher dry season, when assemblages were largely homogeneous, particularly in the arid and nearly hyperarid areas, and only the three sites in the semiarid end of the gradient showed compositional differences. Importantly, the temporal changes in composition at each site also varied in relation to aridity. Seasonal variations in species dissimilarity were lower as aridity increased, regardless of the relative effects of abundance. Similar findings have been reported for ant assemblages along an elevational gradient (Bishop et al., 2014), and corroborate the prediction that seasonal variations in assemblage composition are lower in habitats with more unpredictable climatic conditions (Hawkins et al., 2003). Hence, our results may indicate that desert communities inhabiting highly-unpredictable arid habitats show a higher seasonal similarity than more mesic sites. This is likely due to the ability of some species to cope with the harsh desert conditions regardless of the season (Pierre, 1958; Noy-Meir, 1974; Heatwole, 1996; Ghabbour & Mikhail, 1997).

In contrast, these highly-unpredictable arid habitats showed higher inter-annual compositional variations. However, these inter-annual differences were only significant when accounting for species abundances through the use of Bray–Curtis dissimilarities. This indicates that species composition at each season remains relatively constant from one year to another, and community structure only changes according to the variations in abundance. This is particularly true in the dry season, as revealed by the importance of the Season*Aridity interaction when analyzing inter-annual variations in B–C dissimilarity. Differences in assemblage composition between two consecutive years in arid areas of SE Spain were mainly due to the differences in abundance of a single species, A. baeticus, whose abundance differed one order of magnitude between the two years (Sánchez-Piñero et al., 2011). Large differences between consecutive years in the abundance of particular species are characteristic of Mediterranean arid systems (Noy-Meir, 1974; Sánchez-Piñero et al., 2011). Our results are consistent with the predictions that inter-annual variations in assemblage composition will be higher in more unpredictable habitats (Tonkin et al., 2017), although studies considering a greater number of years will be necessary, especially in the more unpredictable desert sites (Polis, 1991).

Interestingly, changes in aridity through space and time also promote a spatio-temporal shift in the dominance of the two dung beetle subfamilies, Scarabaeinae and Aphodiinae. Aphodiinae are more abundant in the desert communities during the wet season (see also Abdel-Dayem et al., 2016), a change that may be related to a spatial shift in dung use strategies forced by environmental conditions. While all Mediterranean Scarabaeinae species are either paracoprids or telecoprids, the Aphodiinae include endocoprid, saprophagous and kleptocoprid species (Sánchez-Piñero, 1994; González-Megías & Sánchez-Piñero, 2003). Hence, the observed changes in relative abundance of Aphodiinae in the desert communities in the wet season, also reported in previous studies (Labidi, Errouissi & Nouira, 2012; Abdel-Dayem et al., 2016), result in a functional shift from para- and telecoprid beetles to endocoprids and more generalist saprophagous species. Such shift also occurs in time, for Aphodiinae dominate in abundance during the dry season throughout the whole gradient (totaling >95% of individuals in all local communities), a pattern also found in other Mediterranean dung beetle assemblages (Sánchez-Piñero & Ávila, 2004; Sullivan et al., 2016). In fact, our results suggest that the ability of many Aphodiinae species to use different resources for feeding and nesting may allow them to maintain populations in more limiting dry conditions (both in more arid areas as well as in drier seasons), and their relative abundance will diminish when and where milder conditions allow Scarabaeinae to thrive and hold large populations and species-rich communities. Whether this is merely an effect of differences in environmental filtering between the two groups or by the eventual displacement of Scarabaeinae species by the superior competitive ability in dry conditions of the copro-saprophagous Aphodiinae remains an open question.

Conclusions

To summarize, climate-driven environmental filtering is the main process shaping the structure of dung beetle communities along the aridity gradient studied here. However, such filtering may be partly related with the availability of high-quality resources for feeding and nesting. In fact, the diversity–aridity relationship changes through time, determined by the highly variable seasonal and inter-annual patterns of precipitation, that in turn affect the quality and quantity of mammal dung. This results in an ordered replacement of functionally-different species in space and time, as generalist and saprophagous dung beetles become dominant in desert conditions and dryer seasons and years. That is, desert communities are not impoverished subsets of species from species-rich communities from milder climatic conditions, but unique combinations of species adapted to such conditions. Whether this pattern of pre-eminence of species replacement with increasing aridity is common in desert communities needs further investigation since no other studies partitioned beta diversity along aridity gradients before. The limited temporal extent of our study does not allow assessing the effects of large inter-annual changes in precipitation either, but it can, however, be expected that semiarid dung beetle faunas respond to the climate change-driven progressive aridification of East Mediterranean with large functional changes in community structure (see Tonkin et al., 2017). The loss of spatial structure of the dung beetle communities in the dry season for most of the gradient points to a reduction of richness and a higher homogeneity of assemblages if drought becomes a pervasive factor. Indeed, the expected lower seasonal changes in precipitation in progressively more arid conditions is likely to result in a generalized loss of diversity with climate change. But beyond these general patterns of change, the exact nature behind dung beetle responses to aridity remains elusive. Data on functional traits, physiological responses to aridity and long-term community variations are needed to understand the complex mechanisms behind it.

Supplemental Information

Supplemental Information 1 Definitions and summary values of environmental variables used.

Table S1. Summary of grouped variables used (variables used in the final analyses have been marked with a tick mark ✓).

Table S2. Localization (municipality and coordinates), altitude and amount of dung of each replicate and site sampled. Dung availability variables (cow dung and other dung) are presented as an average of the amount found in the four sampling campaigns. Aridity is here presented as the raw original variable, but note that for analysis the inverse of aridity was used in order to have directly interpretable results with increasing aridity.

Click here for additional data file.

Supplemental Information 2 Dung beetle abundance per year (2013 and 2014) and season (wet and dry) along a transect from the Sahara desert to the semiarid Mediterranean coast (see map in Figure 1).

Table S3. Names of species are grouped by subfamily. Totals per site (i.e. community) are given for each subfamily.

Click here for additional data file.

Supplemental Information 3 Supplementary results.

Table S4. Species with significant Indicator Values (IndVal, De Caceres & Legendre, 2009) per year (2013 and 2014) and season (wet and dry).

Table S5. Summary of the GLRs evaluating the effects of the extracted PLS components on species richness and log abundance for the wet and dry seasons.

Figure S1. Variation partitioning of Dung Beetle community variance in raw species composition (based on RDA), beta diversity and Bray-Curtis dissimilarities (based on dbRDA) along the studied aridity gradient.

Click here for additional data file.

We are grateful to Prof. Guy Chavanon for obtaining the sampling permits from the Moroccan authorities, and the Haut Commissariat aux Eaux et Forêts et à la Lutte Contre la Désertification (Direction de la Lutte Contre la Désertification et de la Protection de la Nature) for permission to conduct this research. Pedro Sandoval greatly helped in sorting samples and species identification. Maria de la Luz Tovar and Francisco Martín Peinado (Department of Edaphology, University of Granada) carried out the soil analyses. We thank Daniel Borcard for advice on multivariate variation partitioning and Luis María Carrascal for his invaluable help to conduct PLS-GLR analyses. We thank two anonymous reviewers for their careful reading of our manuscript and their many insightful comments and suggestions. Several researchers, graduate and postgraduate students helped during field campaigns. Thanks to B. Rambert, N. Schriber and M. Julitta for logistic support in Zurich. Finally, special thanks to Elena Cáceres Díaz for motivational support.

Additional Information and Declarations

Competing Interests

Author Contributions

Field Study Permissions

Data Availability

The authors declare that they have no competing interests.

Indradatta deCastro-Arrazola conceived and designed the experiments, performed the experiments, analyzed the data, prepared figures and/or tables, authored or reviewed drafts of the paper, approved the final draft.

Joaquín Hortal conceived and designed the experiments, performed the experiments, authored or reviewed drafts of the paper, approved the final draft.

Marco Moretti contributed to designing the analyses, authored and reviewed drafts of the paper, and approved the final draft.

Francisco Sánchez-Piñero conceived and designed the experiments, performed the experiments, analyzed the data, prepared figures and/or tables, authored or reviewed drafts of the paper, approved the final draft.

The following information was supplied relating to field study approvals (i.e., approving body and any reference numbers):

Field sampling were approved by the Moroccan Authorities and the Haut Commissariat aux Eaux et Forêts et à la Lutte Contre la Désertification (Direction de la Lutte Contre la Désertification et de la Protection de la Nature) (Reference Numbers 01/2013 HCEFLCD/DLCDPN/DPRN/CFF and 01/2014 HCEFLCD/DLCDPN/DPRN/CFF).

The following information was supplied regarding data availability:

The raw data are provided in the Supplemental Files.

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
