# Peer review of "Spatial and temporal variations of aridity shape dung beetle assemblages towards the Sahara desert"

_PeerJ, doi:10.7717/peerj.5210_

## Round 0.1 · original submission · Major Revisions

Both reviewers (and I) found this an interesting and potentially exciting addition to the active research being carried out on dung beetles. Key issue to address is statistical descriptions. Both reviewers have supplied some insightful and important comments that need to be fully addressed in your rebuttal letter.

Also think about the potential of another method: zeta diversity, which has been developed to overcome issues surrounding the use of beta diversity. see Hui C. & McGeoch M. A. (2014) Zeta Diversity as a Concept and Metric That Unifies Incidence-Based Biodiversity Patterns. Am. Nat. 184, 684-94.

and McGeoch M. A., Latombe G., Andrew N. R., Nakagawa S., Nipperess D. A., Roige M., Marzinelli E. M., Campbell A. H., Verges A., Thomas T., Steinberg P. D., Selwood K. E. & Hui C. (2017) The application of zeta diversity as a continuous measure of compositional change in ecology. bioRxiv.

Reviewer 1 ·

Basic reporting

Overall, the writing is good and I enjoyed the manuscript. I have made minor comments throughout. My major concern, however, is over the description of the statistics and analysis. I had great difficulty figuring out what was actually done, and why. I suspect that the analysis is fine, but I cannot properly judge at the moment given the lack of clarity in the methods and results sections. I have suggested improvements that would go some way to fixing this issue.



Line 56-58: OK, so in which ways can the response depend on regions and taxa? I think a couple of brief examples (i.e. "there was a positive/negative response to aridity in these taxa/regions") would help.

Line 72-76: I think you can be a little clearer here. You are saying that Mediterranean zones have hot, dry summers and cold, wet winters. Your next point is that there can still be considerable intra and interannual variation against this "typical" seasonal backdrop, right? At the moment it reads as if this variation is in contradiction to the hot/dry summers and cold/wet winters previously described. Perhaps emphasise that this variation is in addition to the broad seasonal pattern.

Line 74 and 79: You have repeated this exact sentence twice. Consider removing one?
"Importantly, in arid environments the availability of water is unevenly distributed through time, with large and unpredictable variations in rainfall between seasons and years (Ward, 2009)."

Lines 141-145: The explanation of the seasonal differences is very confusing. What are the numbers quoted? The differences between the wet and dry season at particular parts of the gradient? Must be made clearer: e.g "The difference between wet and dry season rainfall (measured as the difference between the wettest and dryest months) was X mm at the coast and Y mm inland."

Figure 1: So is the green-yellow-brown shading the aridity? Please state this in the legend. It should also have a bar showing what green means and what brown means. What are the units of aridity, for example?

Line 171-173: Were the traps separated by 20 m in a straight line? So 80 m separates the 1st and 5th trap? It is not clear if a straight line transect was used or some other design.

Experimental design

The practical methodology is fine and the research questions are well defined and stated in the introduction.

The statistical methods are quite confusing. Would it not be better to arrange them under subtitles (Q1, Q2, Q3) that match your questions and then to have the results presented in a similar way? This would allow these recurring threads to run through the manuscript, and the reader would have a much easier time figuring out which bit of analysis relates to which question.

It is also not clear what has been done with the beta diversity/community composition aspect of the analysis. Multisite indices have been used but in a pairwise fashion…? Why not just use the pairwise indices and use a multiple regression using distance matrices analysis (MRM) to test whether community similarity (measured as turnover or nestedness) was linked to similarity in aridity or any of the other environmental/spatial/temporal factors? MRM are extensions of Mantel tests. See Beaudrot and Marshall (2007, Journal of Animal Ecology, Vol. 80) for use of Mantel and partial Mantel tests. See the MRM function in the R package ecodist, and associated references, for an explanation of MRM techniques.

I am really struggling to understand what kinds of beta diversity have been calculated and how they have been compared. I think you need to break it down and talk the reader through step-by-step. Key to this is state what the unit of analysis is, or, to put it another way, to say what each data point in the corresponding graph would be. I think if you make this information more explicit the entire section would be much easier to read. These beta diversity metrics are tricky, and I cannot properly judge the analysis if I don't know which pairs/groups of sites are being compared to which.

In addition, later on during the environmental correlates section, it seems that point level beta diversity is used. Or, the beta diversity within/between replicates of a given site is used. Is this true? If so it is not clear at all.

There is also a conflation of the terms "species composition" and "beta diversity". In what ways are they different? I would stick to one of terms and use it consistently.

Validity of the findings

The data is high quality. My only reservation is that I cannot properly understand the analyses on beta diversity in the current format of the manuscript.

Additional comments

No other comments.

Reviewer 2 ·

Basic reporting

The document is clear, well written. The introduction introduces well the problem of the structure of dung beetles communities in semi-arid and arid environments, and the assumptions correspond to the problem of the variation of composition along a gradient of aridity. This study can be part of the broader context of an increase in aridity in the countries of North Africa (climate change).
Literature is relevant. However, it would be appropriate to add one reference that completes the subject (Pierre F., 1958). In the cited bibliography, there are also some modifications to make (spelling, name of the editor to complete ...). The list of changes to be made is given below.
Figures are correct. Tables are precise, but another supplementary file is needed (see below for details).

Experimental design

Research questions are relevant, well defined and meaningful. Investigations were performed with ethical standard. All authorizations to collect insects have been obtained from the administrative authorities of Morocco. Statistical analyzes are particularly efficient and satisfactorily used. Beetle collection methods are consistent with those used in similar works. However, it would be desirable that an additional table be provided (in a new supplementary file) giving in particular for each of the 10 inventoried sites their geographic coordinates, and the environmental conditions (in particular the type of soil, and the feces mostly available). This allows in particular to check the validity of some species determinations, such as that of Scarabaeus semipunctatus (cited from site mor. 4), south of the transect, whereas S. semipunctatus is in principle a species of coastal dunes, only in the sand (unless this site n°4 corresponds to the fossil shoreline of an old chott (a chott is a permanent salt water lake with shifting shorelines located in semi-arid regions). The imprecise location makes it impossible to validate this kind of taxonomic data.

Validity of the findings

The results remain interesting and are an important contribution to the knowledge of dung beetle communities in arid and desertic regions. The results and conclusions are well supported by the data analysis. Aridity determines the composition of beetle communities, by filtering species in both space and time. However it is not sure that aridity leads to a functional shift to generalist and saprophagous species. It is not sure the dung beetle species mentioned in Suppl. S2 are saprophagous species. They are dung feeders, and mentioned lines 413-415. In general, detritivorous Scarabaeidae feed on plant residues more or less in putrefaction, which is not compatible with the environmental conditions of semi-desertic sites. This explains it is very important to give the coordinates of the surveyed sites, with the main environmental variables.

Additional comments

The work gives a lot of new data on the organization of dung beetles communities. It is important because very little similar data exists.
However minor corrections are proposed to improve the manuscript..
Line 151: soil
Line 162: prepare a supplementary file on geographic coordinates, as explained previously.
Line 166: forêt
Lines 333-336: need of a short discussion (information) about the dung diversity in sampled sites (cattle vs sheep + other dung categories). The data provided (Appendix 3, fig 1) are too synthetic to provide useful information for understanding the organization of communities.
Line 399: Piera
Line 402: constraints
Lines 420-421: Comments: in dung beetles, the abundance of species depends on the trophic resources (dung) present at the moment of emergence of beetles. When the authors measured the dung density in the sites, did they verify that they counted only fresh or very recent excrements? In arid environments, excrement unused by dung beetles can remain intact for long months, which may overestimate the available trophic resource. What matters is the resource immediately available (see Van Vliet et al. (2009). Factors influencing duiker dung decay in north-east Gabon: are dung beetles hiding duikers? African Journal of Ecology 47(1): 40 – 47). Don’t forget also that often cattle receive in spring veterinary medicinal drugs at moments which can coincide with the time of reproduction of dung beetles. Another possibility: if cattle is scarce (feeding on remnants of crops), this can explain the loss of abundance of dung beetles.
Line 440: remark: it is also the case for Mendidius palmenticola (see Pierre F. (1958). Ecology et peuplement entomologique des sables vifs du Sahara Nord-occidental. Publication du Centre de Recherches Sahariennes, série Biologie, n°1. CNRS édit., Paris).
Line 480: add the reference Pierre, 1958.

References: some changes needed.
Line 642: Institut (with “t”)
Line 705: Revue d’Ecologie et de Biologie du Sol
Line 705: DOI ?? Complete the reference or delete “DOI”
Line 706: Lumaret, J.P.
Line 706: modify the organization of the reference, according the Journal recommendations. Add pagination: 95-115. In Dung beetle Ecology
Line 719; replace “Taylor & Francis” with Blackwell Science, Oxford, U.K.
Line 738: insert her the reference “Pierre, F (1958). See detail of the reference above.
Line 752: add: PhD thesis
Line 765: Gases* (“*” ????). delete
Lines 775-776. Coleopterists Bulletin
Line 785; add; “PhD thesis”


Supplementary S2. Several additions and corrections are needed.
Under the first line “ wet season - Dry season “, insert the numbers of the 10 sites of the transect, in order to make reading easier of the data.

A correction of the spelling of the name Chilothorax is to be made (3 times for the list of 2013, and 3 times for the list of 2014)

Supplementary S3, Table 1
A correction of the spelling of the name Chilothorax is to be made (3 times)

Despite the changes and additions requested, which should improve the work, I consider that it is particularly interesting. The work is well written, the data collected is invaluable. A verification of the identity of a species is necessary (S. semipunctatus). If the determination is correct, it is a major biogeographical contribution for this species (it would be a relictual distribution, a reminder of an ancient period when the level of the Mediterranean Sea was higher than currently).

I suggest a conditional acceptance of the work, after correction and complements.

---

## Round 0.2 · Minor Revisions

Both reviewers and i are happy with your revisions and attention to comments provided. A few minor comments are provided by Reviewer 1.
Thanks again for your revisions.

Reviewer 1 ·

Basic reporting

Again, I enjoyed this manuscript. It is well written, thorough, and the authors have dealt with the comments from myself and the other reviewer well.

Experimental design

Again, this is great.

Validity of the findings

I think the findings are valid.

Additional comments

The manuscript is much improved following the reviews. I have just a few minor points to make. Perhaps just spell out what you mean even more clearly in the text in these few cases.

Line 58: "Diversity patterns of birds are opposite than those for scorpions". But the previous sentence says that scorpion diversity increases with aridity. in America but decreases in Australia. So what is the pattern for birds? Just say what it is without reference to the other taxa.

Line 223: "Significance levels were Bonferroni-corrected to account for multiple comparisons." Are the multiple comparisons you are referring to the temporal pseudoreplication? I.e. the same site is represented multiple times in the dataset as it was sampled in two years and two seasons?

Line 227: So this is Bray-Curtis between each site during each sampling period and every other, correct? Please state.

Multiple-site dissimilarities: so you have computed the multi-site beta among all sites of the gradient within a given year and season? Is this right?

Line 258: This is confusing because I thought you had already described a comparison between seasons and years for each site in line 230 with a PERMANOVA? In which case, what are you describing in line 230? This is still not much clearer from the first version of the manuscript. State what you want to gain from this analysis and the one in line 230.

Reviewer 2 ·

Basic reporting

The authors took into account the comments and remarks of reviewers. The answers given to the various points are satisfactory and reflect a real reflection.
I suggest that the revised version be accepted

Experimental design

Modified and completed

Validity of the findings

OK

Additional comments

Your comments are relevant

---

## Round 0.3 · accepted · Accept

Well done on your manuscript - I believe you have addressed all the minor issues raised in the previous review, and also picked up some other issues raised by these comments. I think this work will be a great addition to the field.

#